# Recommending encounters according to the sociodemographic characteristics of patient strata can reduce risks from type 2 diabetes

Han Ye[1]*, Ujjal Kumar Mukherjee[1], Dilip Chhajed[2], Jason Hirsbrunner[3], Collin Roloff[3]

**1** Department of Business Administration, University of Illinois at Urbana-Champaign, Champaign, Illinois, United States of America, **2** Krannert School of Management, Purdue University, West Lafayette, Indiana, United States of America, **3** Christie Clinic, Champaign, Illinois, United States of America

* hanye@illinois.edu

**Data Availability Statement:** There are legal restrictions on sharing the data set. Christie Clinic does not release clinical data or patient information without a written agreement to the party that will

## Abstract

### Objectives

Physician encounters with patients with type 2 diabetes act as motivation for self-management and lifestyle adjustments that are indispensable for diabetes treatment. We elucidate the sociodemographic sources of variation in encounter usage and the impact of encounter usage on glucose control, which can be used to recommend encounter usage for different sociodemographic strata of patients to reduce risks from Type 2 diabetes.

### Data and methods

We analyzed data from a multi-facility clinic in the Midwestern United States on 2124 patients with type 2 diabetes, from 95 ZIP codes. A zero-inflated Poisson model was used to estimate the effects of various ZIP-code level sociodemographic variables on the encounter usage. A multinomial logistic regression model was built to estimate the effects of physical and telephonic encounters on patients' glucose level transitions. Results from the two models were combined in marginal effect analyses.

### Results and conclusions

Conditional on patients' clinical status, demographics, and insurance status, significant inequality in patient encounters exists across ZIP codes with varying sociodemographic characteristics. One additional physical encounter in a six-month period marginally increases the probability of transition from a diabetic state to a pre-diabetic state by 4.3% and from pre-diabetic to the non-diabetic state by 3.2%. Combined marginal effect analyses illustrate that a ZIP code in the lower quartile of high school graduate percentage among all ZIP codes has 1 fewer physical encounter per six months marginally compared to a ZIP code at the upper quartile, which gives 5.4% average increase in the probability of transitioning from pre-diabetic to diabetic. Our results suggest that policymakers can target particular patient groups who may have inadequate encounters to engage in diabetes care, based on

receive it. Data access queries may be directed to Collin Roloff, Director, Information Management & Analytics at Christie Clinic (contact via Croloff@christieclinic.com). The zip-code level sociodemographic variables were obtained from the US Census Bureau (https://data.census.gov/cedsci/).

**Funding:** The author(s) received no specific funding for this work.

**Competing interests:** The authors have declared that no competing interests exist.

their immediate environmental sociodemographic characteristics, and design programs to increase their encounters to achieve better care outcomes.

## Introduction

In the United States, it is estimated that 30.3 million people (9.4% of the U.S. population) had diabetes in 2015 [1]. The incidence rate of diabetes increased with age, and reached 25.2% among those aged 65 or older [1]. The estimated total cost, including direct medical cost and indirect cost caused by loss of productivity, due to diabetes in 2017 was $327 billion [2]. Type 2 diabetes accounts for 90% to 95% of all diabetes cases [1]. The development of type 2 diabetes is correlated with lifestyle factors such as exercise, weight, nutrition, stress, and urbanization. Type 2 diabetes requires long-term continued care, and patients' engagement in the care process is key to disease management [3,4]. Managing type 2 diabetes requires patients to stay informed from doctors about their medical conditions and treatment practice changes, and get educated about how to control glucose levels and deal with potential complications. Additionally, patients need to routinely self-monitor their glucose levels and may need to take medications in a timely manner. The Chronic Care Model proposed by the Institute for Healthcare Improvement, an independent nonprofit organization, identifies productive encounters between prepared healthcare practice teams and informed and activated patients as the central tenet in managing chronic diabetes and reducing the population-level economic and healthcare burden from diabetes [5].

Both physical and telephonic encounters play an important role in engaging patients in the care process for type 2 diabetes [6–8]. During physical encounters, patients and physicians can meet and discuss patients' medical concerns [9–11]. Since type 2 diabetes may progress over time, physicians can order tests and update patients' clinical conditions during each physical encounter and adjust treatment plans. Physical encounters are also great opportunities for raising awareness of disease and self-management of disease. Telephonic encounters are helpful for patients and care providers to communicate with each other, and often play the role of follow-ups of physical encounters for checking health status and the effectiveness of treatment plans, and understanding any concerns or complications from diabetes [12–14].

The scheduling of physical and telephonic encounters largely depends on patients' clinical conditions [15]. Patients with severe conditions (high blood glucose levels) need to be monitored and meet with physicians more frequently, while patients with mild illness see doctors less often [16]. The insurance status of patients also plays a critical role when deciding encounter frequencies and clinical tests [17]. Patients who pay less out of their pocket for each encounter may tend to schedule more visits in a fixed time period. Moreover, scheduling an appointment is not the same as the actual appointment since patients' adherence to schedule affects the actual number of encounters patients receive. The adherence to schedule may vary from patient to patient.

According to American Diabetes Association, blood sugar levels can be measured by Hemoglobin A1c (A1C), Fasting Plasma Glucose (FPG), Oral Glucose Tolerance Test (OGTT), etc. The current medical diagnosis standard classifies a patient's glucose level into three states: Diabetic (A1C $\geq$ 6.5%; FPG$\geq$126 mg/dl; OGTT$\geq$200 mg/dl), Pre-diabetic (5.7%$\leq$A1C $<$ 6.5%; 100mg/dl$\leq$FPG$<$126 mg/dl; 140 mg/dl$\leq$OGTT$<$200 mg/dl), and Normal (A1C$<$5.7%; FPG$<$100 mg/dl; OGTT$<$140 mg/dl). The glucose level of a patient with type 2 diabetes may fall in any of the three states at a time point, depending on various factors such as self-management and efficacy of treatments. The change in a patient's glucose value

over a time period is often used as a measure of glucose control [18–20]. The transition among the three states of glucose over time periods of a patient reflects disease progress and the effectiveness of disease management. For instance, a transition from Diabetic to Normal in a six-month time period indicates effective disease control, while a transition from Normal to Diabetic indicates disease worsening.

To treat type 2 diabetes, a better understanding of the effects of physical and telephonic encounters on glycemic control (measured by transitions of glucose state) can help policymakers to more efficiently allocate limited capacity of encounters across different patient populations [7]. Inequality of patient encounter is defined as the variation in the frequency of encounter across patients under similar health conditions. Additionally, investigating what factors contribute to the inequalities of encounters across different patient populations can help healthcare providers target specific groups with an elevated risk of high glucose levels. Increasing encounters among the targeted groups may improve their quality of care significantly.

In this article, we report findings from a study that examined sources of inequalities in physical and telephonic encounters of patients with type 2 diabetes at a multi-facility clinic in Illinois. We build two statistical models to estimate: (1) the effects of patients' environmental sociodemographic variables on their encounter utilizations; (2) the effects of telephonic and physical encounters on patients' glucose transitions. The results from the two models can help policymakers target specific patient groups with insufficient encounters, and better allocate encounter capacity at community health centers, clinics, and hospitals to improve diabetes outcomes. Furthermore, we demonstrate complementarities between physical and telephonic encounters, which can be used to target the specific type of encounters for specific patient groups.

## Study data and methods

The data, collected at Christie Clinic in central Illinois, USA, ranges from 02/01/2013 to 12/21/2015 and includes de-identified electronic medical records (EMR) of 10,235 patients with an ICD-10 code related to Diabetes Mellitus. The data set was assessed on 04/25/2016, after the EMR data were fully anonymized. In the data set, there are two metrics that measure patients' glycemic levels: A1C and FPG. All the patients in the sample had their FPG measure as well as A1C measure. However, FPG measures were taken on patients at regular intervals and the A1C measures, which are more robust, were taken at a less frequent interval. Therefore, we used the FPG measure to model the diabetes status of patients. We used the ICD-10 codes to select the final sample for data analysis. The primary disease codes for all types of diabetes are E08, E09, E10, E11, O24, and E13. However, we removed all patients whose disease codes are E08, E09, or E10, which indicate diabetes due to other underlying conditions (23.3% of patients), drug-induced diabetes (13.1% of patients), and Type I diabetes (4.4% of patients). Furthermore, we removed all ICD codes O24 (pre-natal diabetes) and E13 (miscellaneous condition related diabetes), totaling 8.8% of the patients. Patients with Type II diabetes due to other underlying conditions are not usually chronic, and the diabetes usually subsides after the underlying condition is cured. Furthermore, we are interested in observing the effect of continued and regular encounters, both physical and telephonic, on diabetes outcome. Therefore, we removed all patients who do not have sufficient records in the dataset. For example, many patients joined the system only towards the end of the observation period and do not have sufficient records to warrant analysis. In addition, some patients dropped out in the middle of the observation period due to deaths, mobility, or other personal causes. We retained all patients who had records for more than 80% of the observation timeframe. A total of 29.6% of the data

was removed due to the non-availability of data. A total of 2124 patients (20.8% of total initial record) were retained in the final sample of patients. We checked that patients removed from the data due to unavailability of sufficient observations do not have any systematic gender or ZIP-code based variation. These 20.8% of the patients corresponds to 46.3% of the total number of encounters. The time range of the study is divided into 6-month time periods. For each patient at each time period, the data set includes the following variables: glucose (mg/dl), LDL cholesterol (mg/dl), age (years), gender, number of physical encounters, number of telephonic encounters, health insurance policy, ZIP-code of patient home.

It has been shown in extant research that a patient's sociodemographic characteristics (such as income, education, and race) are correlated with both their engagement as well as healthcare outcome [21–28]. Therefore, it is important to incorporate patients' sociodemographic characteristics in the study, since they can be significant explanatory variables and/or confounders for encounters and healthcare outcomes. However, the clinic does not collect individual patients' socioeconomic information. Indeed, patients' socioeconomic context is rarely asked and documented in healthcare systems due to various privacy concerns [29]. Therefore, we collate each patient's ZIP-code as a proxy, which partly reflects their living environment and socioeconomic status with the ZIP-code level sociodemographic information acquired from the 2015 US Census. The ZIP-code level sociodemographic variables include population, annual income, percentage of high school graduates, percentage of college graduates, and race distribution such as percentage of white and percentage of African American.

We first carry out an exploratory clustering analysis, and then build two statistical models motivated by the clustering results: one identifies factors that result in the inequality of patient encounters, while the other estimates the effect of encounters on patients' glucose transitions. Combining the two models together, healthcare providers can identify patients who are likely to have insufficient encounters, and predict the health implications of such a lack of encounters. We elaborate on the clustering analysis and the two models below.

## Exploratory analysis

We first carry out an exploratory data analysis, in which we use three sets of variables to cluster patients with K-means clustering methods [30]. Clustering of patients or locations have been widely used in clinical and healthcare literature to make broad generalizable observations and analysis [31]. The objective of the clustering analysis is to discover evidence of encounter usage inequality across ZIP codes with varying sociodemographic characteristics, after adjusting for clinical measurements. The clustering process allows easy interpretability of results and understanding broad differences between clusters. In particular, patients are clustered into: (1) two groups based on their ZIP-code level socioeconomic information including income and education, (2) two groups based on their ZIP-code level racial distribution, and (3) four groups based on their individual clinical measurements (glucose and cholesterol). In particular, socioeconomic Cluster-1 has higher income and education level than socioeconomic Cluster-2; Racial Cluster-1 has a higher white percentage and lower African American percentage than Racial Cluster-2. Clinical Cluster-4 has the most severe diabetes condition, clinical Cluster-1 has the mildest diabetes condition, while the other two clinical clusters stay in the middle, with respect to disease severity. The descriptive statistics of the variables included in the study and for these three clustering results are shown in Table 1. Hypothesis tests were performed for the descriptive statistics to initially understand the center and variation of data. Description of the tests and the associated null hypotheses is given in Table 1.

We next show how the distribution of patient encounters varies interactively across socioeconomic clusters and clinical clusters in the left panel of Fig 1. In general, patients in

**Table 1. Data description and cluster descriptions.**

| | Mean / Total | Std. Dev | T-Stat / Chi-sq Stat | p-Value |
|---|---|---|---|---|
| **Total Number of Patients** | 2124 | | | |
| **FPG measure (mg/dl)** | 175.5 | 30.6 | 1.65 (H0: FPG≤125) | 0.0495 |
| | | | 2.47 (H0: FPG≤100) | 0.0068 |
| **LDL Cholesterol (mg/dl)** | 172.1 | 34.8 | 2.07 (H0: LDL≤100) | 0.0192 |
| *Patient Gender* | | | | |
| Male | 50.3% | | Chi-sq = 0.004 (df = 1) | 0.9522 |
| Female | 49.7% | | | |
| *Patient Age (Years)* | 64.02 | 11.14 | | |
| < = 40 Years | 2.01% | | Chi-sq = 33.17 (df = 5) | 3.46E-06 |
| 41–50 Years | 8.57% | | | |
| 51–60 Years | 21.75% | | | |
| 61–70 Years | 32.02% | | | |
| 71–80 Years | 19.87% | | | |
| >80 Years | 15.78% | | | |
| Patient Encounters | | | | |
| Six month encounter (physical) | 3.97 | 6.03 | | |
| Six month encounter (telephonic) | 3.01 | 4.82 | | |
| *Health Insurance* | | | | |
| 13 Groups: Patients / Group | 163.4 | 231.1 | Chi-sq = 3921.4 (df = 12) | < 2.2E-16 |
| *Patient Clusters based on Clinical Conditions* | | | | |
| *4 Clusters (Between SS/Total SS)* | | | 73.9% | |
| **Cluster 1 (Percentage of Observations)** | 24.8% | | | |
| Glucose | 113.9 | 18.6 | | |
| Cholesterol | 145.4 | 21.3 | | |
| **Cluster 2 (Percentage of Observations)** | 26.8% | | | |
| Glucose | 169.2 | 20.9 | Hotelling T^2 = 16756.4 | < 2.2E-16 |
| Cholesterol | 145.1 | 22.8 | (Compared to Cluster 1) | |
| **Cluster 3 (Percentage of Observations)** | 21.2% | | | |
| Glucose | 144.7 | 26.4 | Hotelling T^2 = 15641.2 | < 2.2E-16 |
| Cholesterol | 213.2 | 30.8 | (Compared to Cluster 2) | |
| **Cluster 4 (Percentage of Observations)** | 27.2% | | | |
| Glucose | 261.9 | 46.5 | Hotelling T^2 = 9801.2 | < 2.2E-16 |
| Cholesterol | 190.8 | 52.7 | (Compared to Cluster 3) | |
| **Community (Zip Code) Level Descriptives** | | | | |
| Number of ZIP Codes | 95 | | | |
| Number of Patients / ZIP code | 22.4 | 44.7 | Chi-sq = 8405.1 (df = 94) | < 2.2E-16 |
| Population | 6742 | 9618 | Chi-sq = 1289800 (df = 94) | < 2.2E-16 |
| Annual Income ($) | 55822 | 12695 | | |
| Highschool (%) | 91.02% | 5.62% | | |
| College Graduate (%) | 22.62% | 13.65% | | |
| Race—White (%) | 43.82% | 3.01% | | |
| Race—African American (%) | 6.26% | 4.09% | | |
| *Zip Code Clusters: Socio-economic Clusters* | | | | |
| *2 Clusters (Between SS/Total SS)* | | | 64.1% | |
| **Cluster 1** | 42 | | | |
| Annual Income ($) | 66465 | 9059 | | |
| Highschool (%) | 95.1% | 2.17% | | |

(*Continued*)

**Table 1.** (Continued)

| | Mean / Total | Std. Dev | T-Stat / Chi-sq Stat | p-Value |
|---|---|---|---|---|
| College Graduate (%) | 28.8% | 14.15% | | |
| **Cluster 2** | 53 | | | |
| Annual Income ($) | 47389 | 7918 | Hotelling T^2 = 191.33 | < 2.2E-16 |
| Highschool (%) | 87.8% | 5.37% | | |
| College Graduate (%) | 17.7% | 11.11% | | |
| *Zip Code Clusters: Race Based Cluster* | | | | |
| *2 Clusters (Between SS/Total SS)* | | | 81.4% | |
| **Cluster 1** | 25 | | | |
| White Percentage | 48.70% | 1.84% | | |
| African American Percentage | 0.75% | 1.94% | | |
| **Cluster 2** | 70 | | | |
| White Percentage | 42.10% | 3.31% | Hotelling T^2 = 132.35 | < 2.2E-16 |
| African American Percentage | 8.24% | 4.61% | | |

T-test for a population mean is performed for FPG and LDL Cholesterol (H0 is reported with the test statistic). Chi-square goodness of fit test (H0: discrete uniform distribution) is performed for Patient Gender, Patient Age, Patient Health Insurance, Zip-code number of patients, and Zip-code population. Hotelling's t-square test for independent population mean vectors (H0: two population mean vectors are equal) is performed for cluster mean vectors.

socioeconomic Cluster-2 have fewer physical encounters. In other words, after controlling for patients' clinical status, patients who live in a ZIP-code with lower income and education level exhibit fewer physical encounters for diabetes treatment. The right panel in Fig 1 shows the distribution of physical encounters by different clinical clusters and racial clusters. Patients in the same clinical cluster have fewer physical encounters when they belong to Racial Cluster-2 (with lower white percentage and higher African American percentage). These results clearly indicate potential inequalities in patient encounters across ZIP-codes with varying sociodemographic characteristics, after controlling for patients' clinical status.

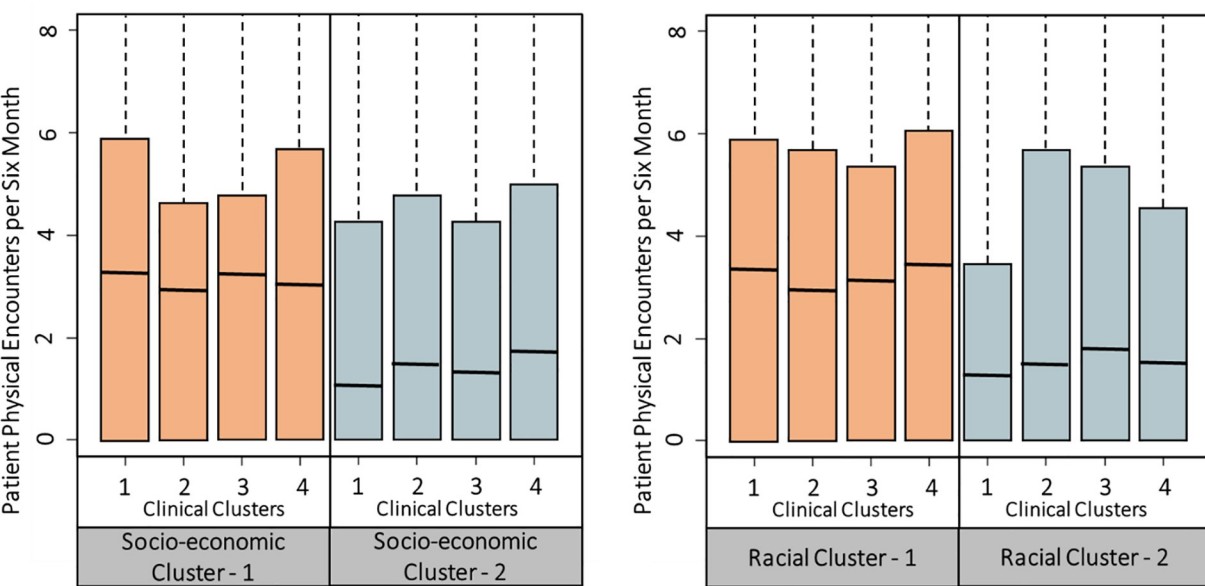

**Fig 1. Significant variation exists in patient encounters based on sociodemographic factors.**

The clustering was done with K-means clustering. The advantage of K-means clustering is that it does not require distributional assumptions and can be done in a non-parametric setting. However, K-means depends on the initial starting points, and therefore tends to suffer from lack of stability in some cases [31]. Therefore, we performed the K-means with 25 random starts and used the most frequent clustering. For the study data, all random starts provided similar clustering results with less than 1% error on average. In addition, we re-estimated the K-means clustering multiple times to ensure stability. Furthermore, in the Appendix we provide a table comparing the clustering by K-means and by Latent Class Analysis (LCA). We find that the differences between the patient clusters are insignificant.

## Model 1: Understanding the inequalities in encounters across ZIP-codes with different sociodemographic characteristics

Motivated by Fig 1, we study how patient encounters vary across ZIP-codes with different sociodemographic characteristics, after controlling for patients' clinical status. Our objective is to identify patient subgroups (e.g., from a ZIP-code with particular sociodemographic characteristics) that are likely to have inadequate encounters for their diabetes control. We build zero-inflated Poisson models [32] for patients' physical and telephonic encounters separately. Patients' six-month count of physical encounters is the response variable in one of the models, while the six-month count of telephonic encounters is the response variable in another model. Poisson regression is often used to model a count response variable. In our data, there is a high frequency of zero counts. Patients may have zero encounter in a six-month period for various reasons such as missed appointments and traveling, which may not be accounted for by the independent variables, and therefore zero counts can be inflated. Hence, we adopt the zero-inflated Poisson model [32] to analyze such zero-inflated data. This model employs two processes to generate count data. One process follows a binomial distribution that generates structural zero counts. The other process follows a Poisson distribution that generates encounter counts given that at least one encounter takes place in a time period. To control for patient clinical status, we include patients' previous time period's clinical measurements of glucose and cholesterol as explanatory variables. Besides, patients' encounters can largely depend on their insurance policy, so we include insurance policy as an explanatory variable. We then fit a zero-inflated Poisson regression model with the Poisson generating process described as follows. The estimated coefficients of the explanatory variables tell their effects on the rate of encounters.

$$log(\lambda_t) = \beta_0 + \beta_1 \, log \, Cholesterol_{t-1} + \beta_2 \, log \, Glucose_{t-1} + \beta_3 \, log \, Age_{t-1} + \beta_4 Gender$$

$$+ \sum_{j=2}^{4} \alpha_j ClinicalCluster_{j,t-1} + \sum_k \tau_k InsuranceType_k + \gamma_1 \, log(population)$$

$$+ \gamma_2 White\% + \gamma_3 AfricanAmerican\% + \gamma_4 \, log(Income) + \gamma_5 HighSchool\%$$

$$+ \gamma_6 College\%.$$

In the equation, $\lambda_t$ is the Poisson rate for the encounter counts at time period $t$. The coefficients $\beta_i$, $\alpha_j$, and $\tau_i$ control for patients' clinical status, insurance policy, and demographics. The coefficients $\gamma_i$ show how ZIP-code level sociodemographic characteristics correlate with patient encounters.

To enhance the robustness of our data analysis, we then fit the above model within each clinical cluster. A different model, namely positive count model, is also fitted for robustness check. The results are essentially similar to that of the zero-inflated Poisson model, and are reported in the appendix.

## Model 2: Estimate the effect of encounters on glucose transition

First, diabetic patients' health status can be majorly reflected by their glucose measurements. Second, encounters may play an important role in controlling diabetic patients' glycemic levels. On the one hand, more encounters mean patients are under improved oversight of their service providers and better informed about their health issues. On the other hand, fewer encounters than needed can mean less attention to healthcare, poorer information, and less education for disease control, and as a result, lead to worse health status. Finally, the ZIP-code level variables record the average values of sociodemographic characteristics of the population around a patient's home, and thus they can partially and indirectly describe the patient's sociodemographic characteristics.

We classify every patient's glucose measurements in each six-month time period into three health states following the common standard: N ($<100$ mg/dl), P (100–125 mg/dl), and D ($>125$ mg/dl). These three states, as mentioned in the introduction, correspond to normal, prediabetes, and diabetes diagnoses, respectively. A patient who is already diagnosed with type 2 diabetes may have their glucose measurement varying among the three states over time. In other words, a patient in any state in time period $t$ may transit to N, P, or D in time period $t+1$. The transition of glucose states tells if a patient's clinical status is improving, worsening, or staying the same. For example, a transition from N to P means a patient's glucose level gets worse, while a transition from D to P means an improvement in glucose level. The chance that a patient in state $i$ (N, P, or D) at time $t$ will transit to state $j$ (N, P, or D) at time $t+1$ can be quantified by a transition probability, denoted by $P_{i \rightarrow j}^t$. The larger the transition probability $P_{i \rightarrow j}^t$, the higher the chance that the patient will transit from state $i$ to state $j$, as time goes from $t$ to $t+1$. Both healthcare providers and patients desire higher probabilities of the transitions that improve the glucose level (e.g., P to N, D to N, and D to P), and lower probabilities of the transitions that worsen the glucose level (e.g., N to P, N to D, and P to D).

The transition probabilities depict how the health risk changes may depend on a patient's engagement via physical and telephonic encounters, as well as the patient's sociodemographic characteristics, which are partially characterized by the ZIP-code level variables. We then build a multinomial logistic regression model [33] to estimate the effect of encounters on glucose status transitions, including ZIP-code level sociodemographic variables and their interactions with the encounter variables. Multinomial logistic regression is a well-developed tool for modeling transitions among a finite number of states, and has been widely adopted in various scientific fields [34–36]. Using the transition of glycemic state as a dependent variable has the advantage of model flexibility such that the effect of an explanatory variable is allowed to vary by different transitions. In contrast, directly using the change in glucose value as a response variable assumes unvaried effect of an explanatory variable, regardless of a patient's current glycemic state. The multinomial logistic regression model for glucose state transitions is stated as follows.

$$\text{Log} \frac{P_{i \rightarrow j}^t}{P_{D \rightarrow D}^t} = \alpha_{0,ij} + \alpha_{1,ij}\text{PhyEncounter}_t + \alpha_{2,ij}\text{TelEncounter}_t + \alpha_{3,ij}\text{logAge} + \alpha_{4,ij}\text{White\%}+$$

$$\alpha_{5,ij}\text{AfricanAmerican\%} + \alpha_{6,ij}\text{log(Income)} + \alpha_{7,ij}\text{HighSchool\%} + \alpha_{8,ij}\text{College\%}+$$

$$\text{PhyEncounter}_t * (\alpha_{9,ij}\text{logAge} + \alpha_{10,ij}\text{White\%} + \alpha_{11,ij}\text{AfricanAmerican\%}+$$

$$\alpha_{12,ij}\text{log(Income)} + \alpha_{13,ij}\text{HighSchool\%} + \alpha_{14,ij}\text{College\%}) + \text{TelEncounter}_t * (\alpha_{15,ij}\text{logAge}+$$

$$\alpha_{16,ij}\text{White\%} + \alpha_{17,ij}\text{AfricanAmerican\%} + \alpha_{18,ij}\text{log(Income)} + \alpha_{19,ij}\text{HighSchool\%}+$$

$$\alpha_{20,ij}\text{College\%}).$$

In the above model, the left-hand side is the logarithm of the odds ratio of the transition from $i$ to $j$ to the transition from $D$ to $D$, where $i, j$ can be any of $\{D, P, N\}$. The coefficient $\alpha_{1,ij}$ indicates the effect of a physical encounter on the log odds of transition from $i$ to $j$. For example, the larger the value of $\alpha_{1,ij}$, the bigger the effect of a physical encounter on the probability of transitioning from $i$ to $j$.

## Study results

### Identify ZIP codes with inadequate encounters

The estimates of Model 1 are shown in Table 2. According to the results, all coefficients show statistical significance when the model is fitted on the entire data to explain physical encounters. Sociodemographic characteristics of a ZIP-code significantly correlate to the number of physical encounters of patients from the ZIP code, after adjusting for patients' clinical measures, gender, age, and insurance policy. Specifically, a smaller population (coef = −0.21, p<0.001), higher percentage of white (coef = 13.71, p<0.001), lower percentage of African Americans (coef = −7.18, p<0.001), higher income (coef = 0.43, p<0.01), higher percentage of high-school graduates (coef = 4.14, p<0.01), and higher percentage of college graduates (coef = 4.59, p<0.001) correlate to a higher number of physical encounters. Besides ZIP-code level variables, other significant factors for predicting physical encounters include dummy variables of clinical clusters 3 and 4 (coef = 0.46, p<0.01; coef = 0.58, p<0.01), log cholesterol in the previous time period (coef = 0.6, p<0.01), log age (coef = 1.96, p<0.001), and male indicator (coef = −0.26, p<0.001).

When the model is fitted within each clinical cluster, the coefficient estimates vary but the general insights remain similar for most ZIP code variables. For clinical Cluster-1 (with lower risk from disease condition) and clinical Cluster-2 (with medium risk from disease condition), the percentage of college graduates (coef = 7.46, p<0.05; coef = 3.13, p<0.01) and log population size (coef = 0.58, p<0.05; coef = 0.35, p<0.001) are both significantly positively related to physical encounters. For clinical Cluster-3 (with medium disease condition), percentage of white (coef = 22.97, p<0.001), and percentage of college graduates (coef = 5.32, p<0.001) are both significantly positively related to physical encounters, while the relationship between the percentage of African American (coef = −11.54, p<0.01) and physical encounters is significant and negative. For clinical Cluster-4 (most severe disease condition), log income (coef = 0.62, p<0.05) and percentage of high-school graduates (coef = 7.9, p<0.05) are the most significant ZIP code factors that correlate to physical encounters, and their effects are both positive. The model estimates show that there exists heterogeneity in physical encounters across ZIP codes with varying sociodemographic characteristics, after adjusting for patients' clinical and demographic factors. The model suggests healthcare providers need to pay special attention to: (i) patients in clinical Cluster-4 (most severe disease condition) from ZIP codes with low-income and low education level, (ii) patients in clinical Cluster-3 from ZIP codes with low education level, low percentage of white, and high percentage of African American, and (iii) patients in clinical Clusters 1 and 2 from ZIP codes with small population and low education level, since these patients are likely to have fewer encounters, which may be inadequate for their disease control.

The results of the model for telephonic encounters are similar to those for physical encounters, showing that the patients from ZIP codes with a larger population, lower-income, and lower education level are likely to have fewer telephonic encounters. Moreover, log age (coef = 1.15, p<0.001) and indicator for clinical Cluster-2 (coef = −0.29, p<0.05) are significantly related to telephonic encounters. The model is also fitted within each clinical cluster. For clinical Cluster-1 (mildest condition), ZIP code variables do not show significant effects

**Table 2. Zero-inflated poisson model estimated for physical and telephonic encounters.**

| Response: Physical Encounters | | | | | | | | | | |
|---|---|---|---|---|---|---|---|---|---|---|
| | All Patients | | Clinical Cluster 1 | | Clinical Cluster 2 | | Clinical Cluster 3 | | Clinical Cluster 4 | |
| **Patient Level** | | | | | | | | | | |
| Log(Cholesterol(t-1)) | 0.6 (0.17, 1.03) | ** | 2.00 (0.18, 3.82) | * | 0.9 (0.21, 1.59) | ** | 0.83 (0.14, 1.52) | * | 1.43 (0.49, 2.37) | ** |
| Log(Glucose(t-1)) | 0.34 (-0.03, 0.71) | + | 1.85 (-0.17, 3.87) | + | 0.63 (-0.21, 1.47) | | 1.02 (0.47, 1.57) | *** | 0.33 (-0.28, 0.94) | |
| Log(PatientAge(t-1)) | 1.96 (1.55, 2.37) | *** | 3.91 (0.87, 6.95) | * | 1.83 (1.14, 2.52) | *** | 2.46 (1.68, 3.24) | *** | 0.97 (0.21, 1.73) | * |
| Patient Gender—Male | -0.26 (-0.4, -0.12) | *** | -0.57 (-1.26, 0.12) | | -0.21 (-0.43, 0.01) | . | -0.3 (-0.52, -0.08) | ** | -0.35 (-0.66, -0.04) | * |
| Clinical Cluster 2 | 0.35 (-0.02, 0.72) | + | | | | | | | | |
| Clinical Cluster 3 | 0.46 (0.13, 0.79) | ** | | | | | | | | |
| Clinical Cluster 4 | 0.58 (0.17, 0.99) | ** | | | | | | | | |
| Insurance | S | | S | | S | | S | | S | |
| **Community Level (Zip Code)** | | | | | | | | | | |
| Log(Population) | -0.21 (-0.33, -0.09) | *** | 0.58 (0.11, 1.05) | * | 0.35 (0.17, 0.53) | *** | 0.14 (-0.04, 0.32) | | 0.14 (-0.15, 0.43) | |
| Percentage White | 13.71 (7.42, 20) | *** | 20.96 (-13.91, 55.83) | | 0.49 (-9.68, 10.66) | | 22.97 (12.66, 33.28) | *** | 8.42 (-6.16, 23) | |
| Percentage African American | -7.18 (-11.32, -3.04) | *** | 13.02 (-9.97, 36.01) | | 2.49 (-4.29, 9.27) | | -11.54 (-18.42, -4.66) | ** | 1.04 (-8.68, 10.76) | |
| Log(Income) | 0.43 (0.16, 0.7) | ** | -1.22 (-2.69, 0.25) | | 0.11 (-0.36, 0.58) | | 0.45 (-0.02, 0.92) | + | 0.62 (0.09, 1.15) | * |
| Percentage High School | 4.14 (1.08, 7.2) | ** | 1.45 (-7.66, 10.56) | | 3.22 (-2.31, 8.75) | | 4.9 (-1.04, 10.84) | | 7.9 (1.53, 14.27) | * |
| Percentage Graduate | 4.58 (3.23, 5.93) | *** | 7.46 (0.72, 14.2) | * | 3.13 (0.93, 5.33) | ** | 5.32 (3.24, 7.4) | *** | 2.75 (-0.86, 6.36) | |
| Number of Patients | 2124 | | 526 | | 569 | | 451 | | 578 | |
| N-Observations | 12533 | | 3104 | | 3357 | | 2661 | | 3411 | |
| Theta (Zero Inflation Factor) | 0.88 | | 0.85 | | 0.94 | | 0.93 | | 0.91 | |
| Log-Lik | -2.96E+04 | | -7.28E+03 | | -9.24E+03 | | -6.72E+03 | | -6.92E+03 | |
| **Response: Telephonic Encounters** | | | | | | | | | | |
| | All Patients | | Clinical Cluster 1 | | Clinical Cluster 2 | | Clinical Cluster 3 | | Clinical Cluster 4 | |
| **Patient Level** | | | | | | | | | | |
| Log(Cholesterol(t-1)) | -0.02 (-0.31, 0.27) | | 0.49 (-0.2, 1.18) | | -0.17 (-0.68, 0.34) | | 0.56 (0.07, 1.05) | * | 0.76 (0.07, 1.45) | * |
| Log(Glucose(t-1)) | -0.01 (-0.26, 0.24) | | -0.25 (-1.13, 0.63) | | 0.23 (-0.44, 0.9) | | -0.22 (-0.61, 0.17) | | 0.25 (-0.2, 0.7) | |
| Log(PatientAge(t-1)) | 1.15 (0.88, 1.42) | *** | 1.27 (0.45, 2.09) | ** | 1.22 (0.71, 1.73) | *** | 1.44 (0.97, 1.91) | *** | 0.65 (0.12, 1.18) | * |
| Patient Gender—Male | 0.06 (-0.04, 0.16) | | 0.21 (-0.12, 0.54) | | 0.07 (-0.11, 0.25) | | 0.03 (-0.13, 0.19) | | 0.01 (-0.19, 0.21) | |
| Clinical Cluster 2 | -0.29 (-0.53, -0.05) | * | | | | | | | | |
| Clinical Cluster 3 | -0.25 (-0.54, 0.04) | + | | | | | | | | |
| Clinical Cluster 4 | 0.23 (-0.02, 0.48) | + | | | | | | | | |
| Insurance | S | | S | | S | | S | | S | |
| **Community Level (Zip Code)** | | | | | | | | | | |
| Log(Population) | -0.11 (-0.19, -0.03) | ** | -0.21 (-0.46, 0.04) | + | -0.28 (-0.42, -0.14) | *** | -0.07 (-0.21, 0.07) | | 0.13 (-0.07, 0.33) | |
| Percentage White | 1.19 (-3.16, 5.54) | | -6.62 (-21.99, 8.75) | | 8.51 (0.63, 16.39) | * | 13.39 (6.14, 20.64) | *** | -5.42 (-14.69, 3.85) | |
| Percentage African American | 0.2 (-2.8, 3.2) | | -1.27 (-11.74, 9.2) | | -3.43 (-8.88, 2.02) | | -6.24 (-11.14, -1.34) | * | 6.08 (-0.39, 12.55) | + |
| Log(Income) | 0.21 (0.01, 0.41) | * | 0.49 (-0.29, 1.27) | | 0.07 (-0.3, 0.44) | | 0.36 (0.01, 0.71) | * | 0.29 (-0.12, 0.7) | |
| Percentage High School | 3.22 (1.63, 4.81) | *** | 3.59 (-1.7, 8.88) | | 3.12 (0.08, 6.16) | * | 2.1 (-0.21, 4.41) | + | 6.28 (1.91, 10.65) | ** |
| Percentage Graduate | 1.85 (1.01, 2.69) | *** | 0.97 (-1.83, 3.77) | | 0.95 (-0.6, 2.5) | | 3.48 (2.15, 4.81) | *** | 0.08 (-1.94, 2.1) | |
| Number of Patients | 2124 | | 526 | | 569 | | 451 | | 578 | |
| N-Observations | 12533 | | 3104 | | 3357 | | 2661 | | 3411 | |
| Theta (Zero Inflation Factor) | 1.21 | | 1.17 | | 1.21 | | 2.01 | | 1.24 | |
| Log-Likelihood | -2.61E+04 | | -6.53E+03 | | -7.16E+03 | | -6.18E+03 | | -7.24E+03 | |

Significance Code. $0 <$ '***' $< = 0.001 <$ '**' $< = 0.01 <$ '*' $< = 0.05 <$ '.' $< = 0.1$.

S: Significant at at least 0.05 level.

Parameter estimates with 95% confidence intervals of the parameter estimates are provided.

on telephonic encounters. For clinical Cluster-2 (medium condition), significant ZIP-code variables include the percentage of white (coef = 8.51, p<0.05), percentage of high school graduates (coef = 3.12, p<0.05), and log population (coef = −0.28, p<0.001). For clinical Cluster-3 (medium condition), the percentage of white (coef = 13.39, p<0.001), log income (coef = 0.36, p<0.05), and percentage of college graduates (coef = 3.48, p<0.001) all have a significant positive effect, while the percentage of African American (coef = −6.24, p<0.05) has a significant negative effect on telephonic encounters. For clinical Cluster-4 (severe condition), percentage of high-school graduates (coef = 6.28, p<0.01) is the only significant ZIP code variable. Therefore, like physical encounters, we observe similar disparity in telephonic encounters across ZIP codes with varying sociodemographic characteristics, after adjusting for patients' clinical and demographic factors. In general, the results can help identify ZIP codes that may correlate to insufficient telephonic encounters (e.g., with high population, low percentage of white, high percentage of African American, low income, and low education level).

We also compute the marginal effect of the sociodemographic variables on patient encounters. As an illustration, we divide the patient populations based on the percentage of high school graduates. We find that patients from a ZIP code at the lower quartile of high school education have 1 fewer average encounter than those at the upper quartile. A similar analysis for income reveals that there are 1.3 fewer average encounters of patients in a ZIP code at the lower quartile than those at the upper quartile.

## Effect of encounters on glucose transition

Estimates of Model 2 are shown in Table 3. Significance level and 95% confidence interval (CI) are also reported for each estimate. The first column denotes the transition of glucose level between two successive time periods. The three states of glucose level N, P, and D are going from mild to severe. Higher probabilities of transitioning to a milder state (e.g., D to P, D to N, and P to N) and lower probabilities of transitioning to a more severe state (e.g., N to P, N to D, and P to D) are desired by both healthcare providers and patients. According to the estimates, more physical encounters predict a higher probability of transitioning from D to P (coef = 0.01, CI: 0.00, 0.02), N to N (coef = 0.02, CI: 0.00, 0.04), and P to N (coef = 0.01, CI: 0.00, 0.02). The results indicate that more physical encounters, after controlling for sociodemographic factors (ZIP-code level), help patients to transit to a milder state from D to P, or P to N, or maintain a healthy glucose level (N to N). Telephonic encounters have a significant negative effect on transition probabilities of N to P (coef = −0.03, CI: −0.05, −0.01) and P to D (coef = −0.02, CI: −0.04, 0.00). It means that more telephonic encounters can help reduce the chance of transitioning to a higher-risk state (N to P and P to D), after controlling for patients' sociodemographic factors (at ZIP-code level).

The interaction coefficient (inter coef) estimates (in Table 3) show how patients' sociodemographic characteristics at the ZIP-code level interact with encounters to influence glucose state transitions. For example, physical encounters have significantly stronger effects on older patients in terms of improving patients' state from D to P (inter coef = 0.08, CI: 0.02, 0.14), keeping patients in state N (inter coef = 0.02, CI: 0.01, 0.03), and preventing transitions from N to D (inter coef = −0.07, CI: −0.13, −0.01) and from N to P (inter coef = −0.17, CI: −0.29, −0.05). While we observe significant racial disparity in encounter usage, the interaction effects on transition probabilities are similar across races in terms of sign, magnitude, and significance level. Other ZIP-code level variables such as income and percentage of college graduates also have significant interactions with physical encounters on some of the transition types. In general, the significant interactions between physical encounters and the ZIP-code variables indicate that increasing encounters would have higher benefits for patients who are older,

**Table 3. Multinomial logistic model estimates for glucose state transition of patients.**

| | Physical Encounter | Telephonic Encounter | Log(Patient Age) | Percentage White | Percentage African American | Log(Income) | HighSchool | Graduate |
|---|---|---|---|---|---|---|---|---|
| | **Main Effects Model** | | | | | | | |
| D→N | 0.00 (-0.02, 0.02) | 0.02 (-0.02, 0.06) | -0.24 (-0.61, 0.13) | 1.99 (-1.02, 5.00) | -4.68* (-7.14, -2.22) | 0.18* (0.04, 0.32) | 0.59 (-0.24, 1.42) | -0.42 (-0.99, 0.15) |
| D→P | 0.01* (0.00, 0.02) | -0.01 (-0.03, 0.01) | 0.18 (-0.08, 0.44) | 3.62* (1.10, 6.14) | -4.16* (-8.16, -0.16) | 0.09* (0.05, 0.13) | 0.31* (0.09, 0.53) | 0.93* (0.52, 1.34) |
| N→D | -0.01 (-0.03, 0.01) | 0.03 (-0.29, 0.35) | 0.42* (0.01, 0.83) | -6.31* (-8.83, -3.79) | 1.68 (-0.82, 4.18) | 0.02 (-0.01, 0.05) | -0.49* (-0.90, -0.08) | -0.74* (-1.41, -0.07) |
| N→N | 0.02* (0.00, 0.04) | 0.02 (-0.02, 0.06) | 0.54 (-0.19, 1.27) | 5.78* (1.13, 10.43) | 0.07 (-0.11, 0.25) | 0.04* (0.01, 0.07) | 4.91* (0.91, 8.91) | 0.19 (-0.14, 0.52) |
| N→P | 0.02 (-0.02, 0.06) | -0.03* (-0.05, -0.01) | 0.13 (-0.11, 0.37) | -1.41* (-2.67, -0.15) | 9.63* (0.98, 18.28) | -0.27* (-0.51, -0.03) | -2.70* (-4.73, -0.67) | -1.56* (-2.47, -0.65) |
| P→D | 0.01* (0.00, 0.02) | -0.02* (-0.04, 0.00) | 0.67* (0.04, 1.30) | -2.58* (-4.96, -0.20) | 4.12* (0.02, 8.24) | -0.10* (-0.16, -0.04) | -0.17 (-0.39, 0.05) | -0.79* (-1.54, -0.04) |
| P→N | 0.01* (0.00, 0.02) | 0.02 (-0.02, 0.06) | -0.13* (-0.23, -0.03) | 4.04* (0.08, 8.00) | -2.34* (-4.39, -0.29) | 0.10 (-0.16, 0.36) | 1.55 (-0.38, 3.48) | 1.05* (0.58, 1.52) |
| P→P | 0.02 (-0.02, 0.06) | -0.04 (-0.10, 0.02) | 0.30 (-0.19, 0.79) | 3.16 (-2.45, 8.77) | 0.03* (0.01, 0.05) | -0.21* (-0.37, -0.05) | -1.49* (-2.49, -0.49) | -0.14 (-0.36, 0.08) |
| | **Interaction Effects** | | | | | | | |
| | *Interaction with Physical Encounter* | | | | | | | |
| D→N | | | 0.08 (-0.04, 0.20) | 1.34 (-0.45, 3.13) | 2.02 (-1.27, 5.31) | -0.01 (-0.03, 0.01) | 0.37* (0.05, 0.69) | 0.18 (-0.06, 0.42) |
| D→P | | | 0.08* (0.02, 0.14) | 1.32 (-0.39, 3.03) | 2.12* (0.13, 4.11) | -0.02 (-0.05, 0.01) | 0.31* (0.05, 0.57) | 0.17* (0.05, 0.29) |
| N→D | | | -0.07* (-0.13, -0.01) | -0.94 (-2.22, 0.34) | -1.61* (-3.01, -0.21) | -0.06* (-0.10, -0.02) | -0.37* (-0.69, -0.05) | 0.02 (-0.01, 0.05) |
| N→N | | | 0.02* (0.01, 0.03) | 1.86* (0.84, 2.88) | 1.71* (0.11, 3.31) | 0.01* (0.00, 0.02) | 0.76* (0.25, 1.27) | 0.09* (0.02, 0.16) |
| N→P | | | -0.17* (-0.29, -0.05) | -1.57* (-2.97, -0.17) | -2.10* (-4.12, -0.08) | -0.13* (-0.22, -0.04) | -0.46 (-1.00, 0.08) | 0.05 (-0.03, 0.13) |
| P→D | | | 0.06 (-0.02, 0.14) | -1.69* (-2.73, -0.65) | -1.66* (-2.86, -0.46) | -0.04* (-0.07, -0.01) | -0.16 (-0.50, 0.18) | -0.12* (-0.22, -0.02) |
| P→N | | | -0.04 (-0.10, 0.02) | 1.99* (0.97, 3.01) | 1.86* (0.43, 3.29) | 0.00 (-0.02, 0.02) | -0.45* (-0.77, -0.13) | 0.05 (-0.04, 0.14) |
| P→P | | | 0.03 (-0.02, 0.08) | 2.59 (-1.22, 6.40) | 2.49 (-1.50, 6.48) | -0.06 (-0.17, 0.05) | -0.30 (-1.04, 0.44) | 0.16 (-0.12, 0.44) |
| | *Interaction with Telephonic Encounters* | | | | | | | |
| D→N | | | -0.18 (-0.52, 0.16) | -0.21 (-0.71, 0.29) | 0.63* (0.00, 1.26) | 0.08 (-0.04, 0.20) | 0.29 (-0.05, 0.63) | 0.03 (-0.02, 0.08) |
| D→P | | | 0.10* (0.04, 0.16) | 1.67* (0.76, 2.58) | 1.90* (0.83, 2.97) | 0.00 (-0.02, 0.02) | 0.21* (0.07, 0.35) | 0.29* (0.09, 0.49) |
| N→D | | | -0.13 (-0.26, 0.00) | -0.67 (-2.12, 0.78) | -0.70 (-1.95, 0.55) | -0.01* (-0.02, 0.00) | -0.06 (-0.16, 0.04) | -0.08 (-0.22, 0.06) |
| N→N | | | 0.00 (-0.03, 0.03) | 0.59 (-0.13, 1.31) | 1.32* (0.27, 2.37) | 0.06* (0.02, 0.10) | 0.33* (0.05, 0.61) | 0.42* (0.04, 0.80) |
| N→P | | | -0.25 (-0.47, -0.03) | -1.03 (-2.85, 0.79) | -1.84* (-3.29, -0.39) | 0.03 (-0.02, 0.08) | -0.42* (-0.80, -0.04) | 0.18 (-0.08, 0.44) |
| P→D | | | -0.10* (-0.19, -0.01) | 0.90 (-0.75, 2.55) | -1.22* (-2.35, -0.09) | 0.02 (-0.01, 0.05) | -0.10 (-0.24, 0.04) | 0.17 (-0.01, 0.35) |
| P→N | | | 0.15* (0.01, 0.29) | 1.14* (0.19, 2.09) | 2.33* (0.16, 4.50) | 0.03 (-0.01, 0.07) | -0.14 (-0.46, 0.18) | 0.32 (-0.06, 0.70) |

*(Continued)*

**Table 3.** (Continued)

| | | | | | | | | |
|---|---|---|---|---|---|---|---|---|
| **P→P** | | | 0.03 (-0.01, 0.07) | -0.39 (-0.93, 0.15) | 0.60 (-0.35, 1.55) | 0.14* (0.03, 0.25) | 0.69* (0.26, 1.12) | 0.31* (0.10, 0.52) |

Notes. 1. D: high glucose, P: medium glucose, N: low glucose.

2. The class transitions likelihoods are estimated using a Multinomial logistic model.

3. The numbers signify multinomial slope estimates.

4. * indicates significant at 0.05 level.

reside in a ZIP-code with a higher percentage of white, a higher percentage of African American, higher income, or higher education level. The only counterintuitive exception is the interaction effect between physical encounters and percentage of high-school graduates on the transition from P to N, which is significantly negative (inter coef = −0.45, CI: −0.77, −0.13), indicating that for patients in ZIP codes with a higher percentage of high school graduates, the effect of physical encounters is smaller in helping the transition from P to N. This is surprising since in general education should be positively associated with lower risk from diabetes. However, we feel that the causal chain is through the nature of patients' occupations, which may be generally less physical for high-school graduates than for non-high-school graduates, and higher levels of physical activity are associated with lower diabetic risks [37]. The current data does not include occupational information of patients.

Telephonic encounters also interact with patients' age and ZIP-code level sociodemographic variables to affect glucose transitions. For example, for older patients, the effect of telephonic encounters is significantly stronger on improving patients' states from D to P (inter coef = 0.1, CI: 0.04, 0.16) and from P to N (inter coef = 0.15, CI: 0.01, 0.29), and preventing state worsening from N to D (inter coef = −0.13, CI: −0.26, 0.00), N to P (inter coef = −0.25, CI: −0.47, −0.03), and P to D (inter coef = −0.1, CI: −0.19, −0.01). For the race variables at the ZIP code level, the interaction between percentage of white and telephonic encounters is significant for two transitions (positive inter coef for D to P and P to N), while the interaction between percentage of African American and telephonic encounters is significant for six transitions (positive inter coef for D to N, D to P, N to N, and P to N; negative inter coef for N to P and P to D). This shows that increased telephonic encounters benefit more in the ZIP codes with higher percentage of either white or African American. In addition, the ZIP codes with high percentage of African American benefits from increased telephonic encounters for more transition types, compared to the ZIP codes with high percentage of white. In summary of the interaction effects, increasing telephonic encounters lead to more benefits among patients who are older, or live in a ZIP code with higher percentage of white, higher percentage of African American, higher income, higher percentage of high-school graduates, or higher percentage of college graduates.

We further interpret the results by computing the average marginal effects of the encounters and sociodemographic variables by computing the marginal effects at each observation in the sample, and averaging the marginal effects for all observations. As an illustration, consider the logistic regression model

$$p_j = \frac{e^{X\beta_j}}{1 + \sum_{j'=1}^{J-1} e^{X\beta_{j'}}}, j = 1, \ldots, J-1,$$

$p_J = \frac{1}{1+\sum_{j'=1}^{J-1} e^{X\beta_{j'}}}$, where $j = 1, \ldots, J$, denote the transitions, $X = (x_1, \ldots, x_K)$ are the values of the $K$ independent variables, and $\beta_j = (\beta_{j_1}, \ldots, \beta_{jK})$ are the coefficients of the $K$ independent

variables for transition category $j$. For a variable $x_k$, its marginal effect for transition category $j$ is given by

$$\left[\frac{\partial p_j}{\partial x_k}\right]_{X=X'} = \frac{e^{X'\beta_j}\left(\beta_{jk} + \beta_{jk}\sum_{j'=1}^{J-1}e^{X'\beta_{j'}} + \sum_{j'=1}^{J-1}\beta_{j'k}e^{X'\beta_{j'}}\right)}{\left(1 + \sum_{j'=1}^{J-1}e^{X'\beta_{j'}}\right)^2}, j = 1,\ldots,J-1,$$

$$\left[\frac{\partial p_J}{\partial x_k}\right]_{X=X'} = \frac{-\sum_{j'=1}^{J-1}\beta_{j'k}e^{X'\beta_{j'}}}{\left(1 + \sum_{j'=1}^{J-1}e^{X'\beta_{j'}}\right)^2}.$$

Then, the average marginal effect of variable $x_k$ on transition $j$ is $\frac{1}{N}\sum_{n=1}^{N}\left[\frac{\partial p_j}{\partial x_k}\right]_{X=X_n}$, where $X_n = (x_{n1},\ldots,x_{nK})$ denotes the $n^{th}$ individual in the data. We used the R package *margins* (https://cran.r-project.org/web/packages/margins/) to compute the average marginal effects of the multinomial transition models. The average marginal effect of physical encounters for all patients for the transition from diabetic to the pre-diabetic stage is 4.3%. This indicates that one additional encounter on average would increase the likelihood of patients' transition D to P by 4.3%. Similarly, the average marginal effect of one additional physical encounter for the transition P to N is 3.2%. This indicates that encounters with doctors, and nurses increase the likelihood of transition from a higher risk level to a lower risk level. To interpret the effect of encounters for different characteristics of patient populations we compute the average marginal effects for different quartiles of patients. The average marginal effect of one additional physical encounter on the transition from diabetic to pre-diabetic for patient groups falling in the upper quartile of the percentage of high school education is 1.1%, while the corresponding average marginal effect for patient groups falling in the lower quartile of the percentage of high school education is 7.9%. This indicates that the effect of encounters for patients with lower levels of high school education is higher than for patients with higher levels. A similar observation is made for the median income and percentage of college graduates.

These results not only identify significant variation in the effect of physical and telephonic encounters on the health risks from diabetes across patient groups, but also indicate that physical encounters work better for improving disease states (such as increasing chances of the good transitions: D to P, N to N, and P to N), while telephonic encounters work better for preventing disease state worsening (such as reducing chances of the bad transitions: N to P, and P to D). These results, therefore, indicate the prevalence of complementarities between the two types of encounters, which can be used by providers to focus strategically on specific encounter types.

We also combine the results of the two models to interpret the effects of sociodemographic factors on encounters, and the effect of the encounters on the final diabetes outcome. As an illustrative example, we observe that for ZIP-codes which are in the lower quartile of the distribution on the variable percentage of high school graduates, there is a marginal Poisson rate of 1 fewer encounter (from zero-inflated Poisson models) compared to the ZIP-codes at the upper quartile of percentage of high school graduates, which when plugged into the multinomial regression model gives an 5.4% average increase in the probability of transitioning from pre-diabetic to diabetic. The corresponding estimates for median zip code income are 1.3 and 7.3%, respectively.

## Discussion

Inequalities in healthcare outcome often stem from disparities in access to healthcare resources, such as encounters. We show inequalities exist in physical and telephonic

encounters across patients' ZIP-codes with heterogeneous sociodemographic characteristics, after controlling for patients' clinical, demographic, and insurance variables. We also show the implication of physical and telephonic encounters for patients' glucose transition. Furthermore, we demonstrate that the two encounter types act differently on patient subpopulations with different sociodemographic status. Therefore, the two encounter types can be used in a complementary manner, differentially on different patient subpopulations to improve overall risk levels from diabetes under capacity constraints of patient encounters, especially when there is an increasing push towards more inclusiveness of patients from all sociodemographic backgrounds into the purview of managed care under government regulatory and legislative initiatives.

The results may help healthcare providers to target patient subgroups from specific ZIP-codes according to ZIP-code sociodemographic characteristics, who may have inadequate encounters to engage in diabetes care. Interventions are supported by extant research to target those patients with inadequate encounters [38]. Governmental and social support in terms of providing better access to healthcare resources, particularly preventative and primary care resources is important in improving overall healthcare measure and reducing healthcare inequality. Programs to engage targeted patient groups may be designed for improving encounters and engagement. For instance, education events may be provided in the ZIP-codes with inadequate encounters to raise the awareness of disease prevention and active control; nurses and case managers may arrange more telephonic encounters for consultation, education, appointment reminders, etc.

Since the total encounter capacity is often limited, increasing encounters among particular patient subgroups means that the number of encounters of some other patients needs to be reduced. Model 1 can be used to identify patient subgroups who have a high number of encounters, after controlling for their clinical status. Appropriately reducing encounters for the patients who have more than sufficient encounters may not have negative effects on their diabetes control. Indeed, proper reallocation of encounters can help reduce the inequality in limited healthcare resources, and lead to better outcomes of the entire patient population. In order to address such inequality in encounters, especially in physical encounters, telemedicine and online communication can be used to reduce the burden of the "super-utilizers" [39,40].

Patients' socioeconomic contexts have been shown in extant research to be significant explanatory variables or predictors for healthcare outcome and disease risks [41,42]. Nevertheless, healthcare providers have seldom collected such information. There are concerns about the value, feasibility, and efficiency of collecting such patient data at the individual level [29]. Our study supplements the unavailable individual socioeconomic data with the ZIP-code socioeconomic data from the US census, and shows that it explains a significant amount of variation in patient encounter utilizations as well as glucose control outcomes. This supports the use of population data from geographic areas to infer individual patients' socioeconomic context, and can be helpful to other studies that may need yet lack individual patient socioeconomic data.

Although the study is based on the data from a regional clinic in Illinois, the methods can be carried over to analyze data collected at other clinics as well. The insights from the study may be carried over to other regions where patients share similar sociodemographic and clinical features. Besides identifying patient subgroups based on ZIP-codes that require more encounters to enhance engagement, the models 1 and 2 can also be used to predict glucose measurement and encounter utilization for individual patients. Engagement programs may be tailored to individual patients to achieve the highest efficiency in encounter utilization and reduce the inequality in encounters and healthcare outcomes.

## Conclusion

In closing, there exist inequalities in physical and telephonic encounter utilizations across ZIP-code areas with varying sociodemographic characteristics, after adjusting for patients' clinical status, demographics, and insurance policy. The inequalities in encounter utilizations may lead to disparity in diabetes care outcomes. Policymakers should consider actions such as increasing healthcare capacity or designing programs for targeted patient groups with inadequate encounters to mitigate such inequality in encounter utilization, and ultimately improve the efficiency of care and the healthcare outcome of the entire served population. Also, healthcare providers and policymakers have the opportunity to consider complementarities between encounter types while planning chronic disease care which requires continued and repeated encounters of patients with the healthcare systems. The results of this study demonstrate the importance of designing healthcare systems that are able to recommend encounter frequency to the type for patient subpopulations characterized by specific sociodemographic characteristics.

## Supporting information

**S1 Appendix. Estimates for positive count model of physical and telephonic encounters.**
(DOCX)

## Author Contributions

**Conceptualization:** Han Ye, Ujjal Kumar Mukherjee, Dilip Chhajed, Jason Hirsbrunner, Collin Roloff.

**Data curation:** Jason Hirsbrunner, Collin Roloff.

**Formal analysis:** Han Ye, Ujjal Kumar Mukherjee, Dilip Chhajed.

**Methodology:** Han Ye, Ujjal Kumar Mukherjee, Dilip Chhajed.

**Writing – original draft:** Han Ye, Ujjal Kumar Mukherjee, Dilip Chhajed.

**Writing – review & editing:** Han Ye, Ujjal Kumar Mukherjee, Dilip Chhajed.

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
