## [Decision Letter · Decision Letter 0]

23 Nov 2020

PONE-D-20-27788

Customizing Patient Encounters for Different Socioeconomic and Demographic Strata Can Reduce Risks from Type 2 Diabetes

PLOS ONE

Dear Dr. Ye,

Thank you for submitting your manuscript to PLOS ONE. After careful consideration, we feel that it has merit but does not fully meet PLOS ONE’s publication criteria as it currently stands. Therefore, we invite you to submit a revised version of the manuscript that addresses the points raised during the review process.

We look forward to receiving your revised manuscript.

Kind regards,

Antonio Palazón-Bru, PhD

Academic Editor

PLOS ONE

Journal Requirements:

2.Thank you for your submission to PLOS ONE.

We note that your manuscript states that your IRB approved this research, however your ethics statement states that the research activities described in this application meet the criteria for exemption at 45CFR46.101(b)(4). Please clarify whether the IRB approved your study or whether you obtained a waiver from ethics approval from the IRB due to exempt status.

In addition, please clarify whether exemption was for ethics approval and informed consent, or whether written informed consent was obtained from subjects.

Please also provide the name of the multi-facility clinic in central Illinois.

Also, please include the date(s) on which you accessed the databases or records to obtain the data used in your study.

3.We note that you have indicated that data from this study are available upon request. PLOS only allows data to be available upon request if there are legal or ethical restrictions on sharing data publicly. For information on unacceptable data access restrictions, please see http://journals.plos.org/plosone/s/data-availability#loc-unacceptable-data-access-restrictions.

Reviewers' comments:

Reviewer's Responses to Questions

**Comments to the Author**

1. Is the manuscript technically sound, and do the data support the conclusions?

Reviewer #1: No

Reviewer #2: Yes

2. Has the statistical analysis been performed appropriately and rigorously? 

Reviewer #1: No

Reviewer #2: Yes

3. Have the authors made all data underlying the findings in their manuscript fully available?

Reviewer #1: No

Reviewer #2: Yes

4. Is the manuscript presented in an intelligible fashion and written in standard English?

Reviewer #1: No

Reviewer #2: Yes

5. Review Comments to the Author

Reviewer #1: My thanks to the editor and authors for the opportunity to review this manuscript. This manuscript reports the results of a retrospective study of the effect of social and demographic variables on blood glucose among persons with diabetes. It is an important topic and this manuscript fills an important gap in the research literature. Below I provide some comments and suggestions to the editor and authors that also serve as rationale for my recommendation. Overall, I do not think the manuscript is appropriate for publication but the authors might consider a separate and new analysis taking into account some of these critiques.

Minor: I do not see what “customizing” has to do with the hypothesis being tested in this manuscript. This study does not seem to be about customization, but rather recommendations for care utilization patterns.

Abstract:

• The abstract begins with a description of the importance of physician encounters and “the inverse law of patient encounters” which is seems to be mostly conventional wisdom rather than established theory. I recommend removing the first to sentences and providing a short and clear description of the rationale for the paper, grounded in literature on socioeconomic status and health care utilization. A more straightforward statement of the primary hypothesis would be clearer than the current presentation in the abstract.

• The abstract includes unnecessary stylistic language (e.g. “last but not least”) that detracts from the message.

• Description of “do correlate” should be accompanied be correlation coefficient or other statistic that illustrates this finding.

• The abstract seems to confuse segregation (measured as African Americans in a zip code) with socioeconomic status. The authors need to be accurate in their description of concepts.

Manuscript

• The body of the manuscript has some grammar problems. I have only mentioned the first few to appear. The authors should edit and revise their work more carefully. For example:

o Pg 3 ln. 60 is improperly worded. The prevalence did not increase with age, the prevalence is higher among older persons.

o Pg. 3 ln 62 “in 2017 is $327 billion” should be “was”

• Pg 5 ln 103 “glucose transitions” is not defined or referenced until much later in the paper and most readers will not understand what is being referred to here. The transition approach is elaborate and does not appear to have been reported or described elsewhere. The authors likely need to establish the validity of this approach in a separate manuscript.

• Pg 5 which IRB approved the study?

• Pg 5 How were the data collected? Chart abstraction? Electronic medical record? What procedures were used?

• Given that patients have A1C values, why use FPG as the outcome instead of A1C?

• Were A1C and FPG point of care or from a venous blood draw in the lab?

• Zip codes are large and potentially immensely heterogeneous. Why not geocode to the census tract or block group to get more fine grained information? Tools like tidycensus and sociome provide the opportunity for higher resolution, neighborhood level analyses that do not suffer from the same degree of weakness as zip codes.

• Major: The manuscript is presented as though all patients had complete data, but in my experience that is rarely the case. How many patients were excluded due to incomplete data?

• Major: The methods section does not appear to describe inclusion/exclusion criteria.

• What are the tests for which p values are reported in Table 1? Are there hypotheses tied to these tests?

• Table 1 refers to “clusters” but insufficent explanation in the table is given about what is meant by “cluster” How did the authors decide to create these clusters?

• The state transition framework for office measurements of FPG is somewhat innovative and potentially useful.

• Too little information is provided about the number and frequency of measurements of FPG per patient.

• It seems that patients with insufficient follow-up visits would be excluded because they would have no transition data. This is a potential limitation of the design. Doesn’t this mean that Model 2 should come first? Wouldn’t the probability of having repeated visits influence the frequency of measurement and thus the potential to observe a glucose transition?

• IT does not seem to make sense that the cluster analysis is embedded in model 2. Should not the cluster analysis be presented as having its own analytic plan and hypotheses? What was the rationale for this step other than data reduction? How do these locally, empirically defined clusters compare to other approaches of transforming zip code and clinical measurement data?

• K-means cluster analysis is notorious for having problems with reproducibility. This is a serious limitation of the analysis. The authors should recognize this limitation and consider or propose future working using more sophisticated techniques like latent class analysis or factor mixture models.

• Major weakness: Detailed results of the cluster analysis are largely omitted from the manuscript.

• There are many comparisons made in Table 2.

• Table 2 should have a sample size. No mention is made of whether there was any adjustment for multiple comparison, drawing into question the validity of any of the bolded “significant” findings. The authors need to address for multiple comparison in this type of table, or if they have done so, should say that they did. Significance tests are a poor and possibly invalid approach here anyhow (see American Statistical Association recent public statements on this matter) and 95% confidence intervals for each estimate would be much preferred.

• Table 2 seems a little bit more like an all by all table of raw output rather than a presentation of results. With so many interaction effects the results presentation is both bewildering and suspect.

• All of the interpretations about results surrounding the larger percentage of African Americans in a zip code are suspect. This is especially concerning given that “Cluster 2” has just 8% African American population. Are we to believe that there is something special about zip codes with 8% vs. 4% African Americans as it relates to diabetes care? What theory in the literature would suggest this is important for us to look at?

• Although there are good reasons to examine socioeconomic status and glucose control, and the authors appear to have assembled a useful regional data resource, the analytic plan is insufficiently described and poorly justified. Critical details are omitted from the manuscript (missing data, inclusion/exclusion criteria). The analytic methods and reported results, especially the clustering, the multiple comparisons in Table 2 and the many inferences made about small observed magnitude differences render the results and discussion unable to convince this reviewer of the accuracy, relevance or salience of any of the study findings.

Reviewer #2: 1. The study design was clearly laid out for the reader. Variables of interest were well defined for the reader.

2. The statistical methods were clearly explained in the paper and appropriate to the research question.

3. The statistical methods were appropriate to analyze the variables of interest and the rationale for the researcher(s) choice(s) of statistical methods was explained.

4. The subject matter was relevant to diabetes care and to more general patient care areas.

5. The subject matter of the study is an important area of concern regarding diabetes care outcomes in varying sociodemographic settings.

6. Allocation of medical care is based on a number of variables. The variables of interest in the study are some of the variables identified in other previous studies that examine the role of social determinants of health in population care outcomes.

7. This study adds to the body of knowledge in the subject area of diabetes care outcomes by identifying another aspect of care models and delivery of health care services to address areas of greater need.

8. Contentions of the author(s) were supported by the data.

The article was a very technical read in terms of data analyses and discussion of results. It contributes further to the body of knowledge in the subject area.

6. PLOS authors have the option to publish the peer review history of their article (what does this mean?). If published, this will include your full peer review and any attached files.

Reviewer #1: No

Reviewer #2: **Yes: **Pamela Phares PhD

---

## [Author Response · Author response to Decision Letter 0]

22 Jan 2021

We have attached a document containing our point-by-point responses to the review team's comments.

---

## [Decision Letter · Decision Letter 1]

11 Mar 2021

Recommending Encounters According to the Sociodemographic Characteristics of Patient Strata Can Reduce Risks from Type 2 Diabetes

PONE-D-20-27788R1

Dear Dr. Ye,

We’re pleased to inform you that your manuscript has been judged scientifically suitable for publication and will be formally accepted for publication once it meets all outstanding technical requirements.

Kind regards,

Antonio Palazón-Bru, PhD

Academic Editor

PLOS ONE

Additional Editor Comments (optional):

All the reviewers' concerns have been correctly addressed.

Reviewers' comments:

Reviewer's Responses to Questions

**Comments to the Author**

1. If the authors have adequately addressed your comments raised in a previous round of review and you feel that this manuscript is now acceptable for publication, you may indicate that here to bypass the “Comments to the Author” section, enter your conflict of interest statement in the “Confidential to Editor” section, and submit your "Accept" recommendation.

Reviewer #2: All comments have been addressed

2. Is the manuscript technically sound, and do the data support the conclusions?

Reviewer #2: Yes

3. Has the statistical analysis been performed appropriately and rigorously? 

Reviewer #2: Yes

4. Have the authors made all data underlying the findings in their manuscript fully available?

Reviewer #2: Yes

5. Is the manuscript presented in an intelligible fashion and written in standard English?

Reviewer #2: Yes

6. Review Comments to the Author

Reviewer #2: 1. Reviewing comments of Reviewer #1 and responses from the authors, they appear to have addressed the problematic issues with Kmeans clustering algorithms, data source, data interpretation and data presentation (from the edited version). Explanations for the weaknesses in Kmeans algorithms and their use of multiple iterations using different starting points as well as using LCA analysis and comparing these results to the Kmeans clustering data appear to address some of the inherent problems with the use of Kmeans algorithms. Use of zip-code level data is commonly used with data sets that are restricted for privacy reasons and are useful to the extent of their limitations.

2. Again, going point by point in their analyses and conclusions I do not see any major errors.

4. Yes, the data appear to be complete except for those that they are not privy to for confidentiality reasons.

5. I think the editorial revisions that I reviewed have made this a more coherent paper with better grammatical composition overall.

I recommend publication of the paper. The major issues brought forth regarding the paper appear to be addressed satisfactorily. The study is a good starting point for additional studies analyzing the relationship among healthcare usage, healthcare outcomes and geographic areas that are underserved by healthcare delivery systems and public health initiatives.

7. PLOS authors have the option to publish the peer review history of their article (what does this mean?). If published, this will include your full peer review and any attached files.

Reviewer #2: No

---

## [Editor Report · Acceptance letter]

15 Mar 2021

PONE-D-20-27788R1 

Recommending Encounters According to the Sociodemographic Characteristics of Patient Strata Can Reduce Risks from Type 2 Diabetes 

Dear Dr. Ye:

I'm pleased to inform you that your manuscript has been deemed suitable for publication in PLOS ONE. Congratulations! Your manuscript is now with our production department. 

Kind regards, 

on behalf of

Dr. Antonio Palazón-Bru 

Academic Editor

PLOS ONE